

# An ensemble approach for research article classification: a case study in artificial intelligence

Min Lu[1], Lie Tang[1] and Xianke Zhou[2]

[1] Hangzhou Science and Technology Information Institute, Hangzhou, Zhejiang, China
[2] Institute of Computer Innovation, Zhejiang University, Hangzhou, Zhejiang, China

## ABSTRACT

Text classification of research articles in emerging fields poses significant challenges due to their complex boundaries, interdisciplinary nature, and rapid evolution. Traditional methods, which rely on manually curated search terms and keyword matching, often lack recall due to the inherent incompleteness of keyword lists. In response to this limitation, this study introduces a deep learning-based ensemble approach that addresses the challenges of article classification in dynamic research areas, using the field of artificial intelligence (AI) as a case study. Our approach included using decision tree, sciBERT and regular expression matching on different fields of the articles, and a support vector machine (SVM) to merge the results from different models. We evaluated the effectiveness of our method on a manually labeled dataset, finding that our combined approach captured around 97% of AI-related articles in the web of science (WoS) *corpus* with a precision of 0.92. This presents a 0.15 increase in F1-score compared with existing search term based approach. Following this, we performed an ablation study to prove that each component in the ensemble model contributes to the overall performance, and that sciBERT outperforms other pre-trained BERT models in this case.

## INTRODUCTION

The total output of scientific research has seen a significant increase in recent years, with global scientific publishing output being 53% higher in 2020 compared to 2010. This growth is even more pronounced in emerging research fields, particularly in areas like artificial intelligence, robotics energy and materials science (*White, 2019*). Emerging research fields, characterized by complex boundaries and rapid evolution (*WIPO, 2019*), present unique challenges for accurate identification of research articles. Despite these difficulties, research on emerging fields holds great importance in understanding the evolution and impact of new technologies and scientific domains. For example, understanding the field of AI is crucial due to its interdisciplinary nature and its significant impact on various sectors ranging from economics to social development. Artificial intelligence (AI), integrating elements from computer science, biology, psychology, and more, plays a pivotal role in advancing technologies like speech recognition, image processing, and intelligent robotics, thereby revolutionizing labor efficiency and creating new job demands (*Zhang & Lu, 2021*). By examining the various aspects of these fields,

Corresponding author
Lie Tang, tanglie23@163.com

researchers and policymakers can gain valuable insights into their growth, trends, and potential implications for society.

The development of AI began in the 1950s, marking significant milestones such as the creation of neural networks and the introduction of terms like 'artificial intelligence' and 'machine learning.' More recent developments in the 2000s include IBM Watson, facial recognition technology, and autonomous vehicles (*Haenlein & Kaplan, 2019*). AI has gained significant attention in recent years, becoming a popular topic across various research fields. AI encompasses a wide range of techniques and methods that allow machines to perform tasks that typically require human intelligence. These tasks include, but are not limited to, problem-solving, learning, perception, and language understanding (*Russell, 2010*). The advancements in AI have led to a plethora of sub-fields, such as machine learning, natural language processing, computer vision, and robotics (*WIPO, 2019*).

AI's impact is widespread across industries. In healthcare, AI contributes to risk management, analytics, and knowledge creation, aiding in complex diagnostics and disease prevention (*Jiang et al., 2017*). In the public sector, AI is used to make cities safer and cleaner, aiding in traffic, pollution, and crime prediction (*de Sousa et al., 2019*). Retail has seen AI applications in predictive analytics and customer service, enhancing operational efficiency and customer experience (*Oosthuizen et al., 2021*). AI's contribution to manufacturing includes research and development, predictive analytics, and real-time operations management, notably impacting industrial automation (*Arinez et al., 2020*). These applications highlight AI's transformative potential in augmenting human abilities and driving significant economic growth. As a result of the growing interest in AI, the volume of AI-related articles has grown exponentially (*Benefo et al., 2022*), highlighting the importance of tools for the accurate identification of AI-related articles. However, accurately identifying AI-related articles within the vast *corpus* of scientific literature is challenging due to the broad, ambiguous, and rapidly evolving nature of the field (*WIPO, 2019*).

In the realm of research article classification, conventional keyword-based searches are a standard approach. For example, in the field of dentistry, a comprehensive review highlights the importance of correctly using keywords and Boolean operators for effective literature searches (*Khurshid et al., 2021*). Similarly, in a study on acute respiratory tract infections (ARTI) and human metapneumovirus (hMPV) in children, researchers used a title and abstract (TIAB) search strategy with major keywords on hMPV infections to extract relevant data from various international publications. This approach helped in analyzing the epidemiological aspects of hMPV infections in children globally (*Divarathna, Rafeek & Noordeen, 2020*). Both examples underscore the significance of keyword-based searches in efficiently sifting through and classifying large volumes of academic literature across diverse fields.

Another conventional method for classifying research articles involves using generic classification systems, such as the Web of Science (WoS) categories and Scopus categories. For instance, a bibliometric analysis in orthopedics utilized the WoS category to identify influential studies in the field, demonstrating the application of these systems for academic

analysis (*Li et al., 2019*). Similarly, a study reviewing journal categories in WoS, Scopus, and MathSciNet bases under the title quartiles illustrated the use of these databases for journal ranking and classification, highlighting the significance of such systems in evaluating and categorizing scientific literature (*Jabbari Nooghabi & Alavian, 2022*). These examples show the prevalence of generic classification systems in scholarly research, offering a structured approach to categorize and evaluate academic publications.

Deep learning techniques have also been increasingly employed in the classification of scholarly articles. A study using convolutional neural networks (CNNs) for scientometric analysis and classification demonstrated this approach's effectiveness. By incorporating publication, author, and content features, both explicit and implicit, into CNN models, this method achieved higher precision, recognition, and F1-score than traditional machine learning methods (*Daradkeh et al., 2022*). Another research utilized transformer models like BERT, specifically SciBERT, for text classification of peer review articles. This study found that sentence embeddings obtained from SciBERT, combined with entity embeddings, significantly enhanced classification performance, demonstrating the potency of deep learning methods in this domain (*Piao, 2021*).

In this study, we propose a novel ensemble approach to improve the classification of research articles in the field of AI. Our contributions include:

(1) We propose a novel process for retrieving search articles related to a given topic. We first retrieve articles from the Web of Science (WoS) based on a combination of search terms, categories, and high-frequency keyword analysis. We then use an ensemble model to further identify the relevant articles.

(2) We develop an ensemble model that combines decision trees, SciBERT, regular expression matching, and support vector machines (SVM) to enhance research article identification accuracy.

(3) We conduct an evaluation of our ensemble approach through ablation studies and cross-validation, demonstrating that each component in the model contributes uniquely to its overall performance.

We manually labelled a set of 4,000 articles as either AI-related or non-AI for training, and these labels were used to calculate the precision and recall of our approach by 5-fold cross-validation. The results yielded a precision of 92% and a recall of 97%.

## LITERATURE REVIEW

### Research article classification in emerging fields

Bibliometric methods are frequently employed in the classification of research articles, particularly in emerging fields. These methods involve a comprehensive analysis of publication patterns and trends. For instance, in synthetic biology, a study outlined a bibliometric approach to define the field by analyzing a core set of articles and refining keywords identified from these sources, also including articles from dedicated journals (*Shapira, Kwon & Youtie, 2017*).

In nano-energy research, *Guan & Liu (2014)* used bibliometric and social network analysis to investigate the exponential growth of research output and compare scientific performances across countries. This study revealed a shift in the global share of nano-

energy research, with emerging economies like China showing significant development momentum. The research also highlighted the evolving patterns of scientific collaboration in this field, marking a shift in influence from traditional scientific powerhouses to emerging economies.

The detection of emerging research topics (ERTs) using bibliometric indicators was the focus of another study (*Xu et al., 2021*). This research distinguished ERTs from common-related topics and developed a method to uncover high-impact ERTs with a fine level of granularity. The study not only identified ERTs but also proposed different research and development strategies for each topic, emphasizing the future economic and social impact of these ERTs and the reduction of uncertainty in research directions.

### Search strategies of artificial intelligence

Previous efforts to classify articles related to AI have employed a variety of strategies. A simple yet commonly used method involves search strategies such as TS=("artificial intelligence"), which is straightforward but may miss nuanced aspects of AI research. For instance, a study examining AI developments in China used this strategy to analyze the interaction between academic research and policy-making, providing insights into the evolving relationship between these domains (*Gao, Huang & Zhang, 2019*).

More complex approaches involve refining and complementing keywords with subject categories. Another study adopted a bibliometric definition for AI, starting with core keywords and specialized journal searches. This method was then enhanced by extracting high-frequency keywords from benchmark records, which was compared with other search strategies to profile AI's growth and diffusion in scientific research. This approach allowed for a more detailed understanding of AI's multidisciplinary development and the contributions from diverse disciplines (*Liu, Shapira & Yue, 2021*).

In addition, the World Intellectual Property Organization (WIPO) in 2019 applied a comprehensive search strategy that combined patent classification codes with an extended list of keywords. This strategy was based on a thorough literature review, established hierarchies, web resources, and manual checking. To identify AI-related publications, approximately 60 words or phrases specific to AI concepts were queried across all subject areas in the Scopus scientific publication database. Additionally, about 35 words or phrases related to AI were applied specifically to the Scopus subject areas of Mathematics, Computer Science, and Engineering (*WIPO, 2019*).

### Deep learning models in research article classification

Deep learning models have also been utilized in identifying AI-related articles and patents. *Dunham, Melot & Murdick (2020)* described a strategy leveraging the arXiv *corpus* to define AI relevance, employing deep learning techniques like convolutional neural networks. This approach yielded high classification scores and precision in identifying AI-relevant subjects, demonstrating the effectiveness of supervised solutions in updating AI-related publication identification in line with research advancements. *Miric, Jia & Huang (2023)* demonstrated the use of machine learning tools,

including deep learning, for classifying unstructured text data. The study focuses on identifying AI technologies in patents, comparing machine learning (ML) methods with traditional keyword-based approaches. The findings indicate the superiority of ML in terms of accuracy and efficiency, underscoring the advantages of these methods in capturing AI-related content in patents. *Siebert et al. (2018)* utilized both supervised and unsupervised machine learning approaches for analyzing a large *corpus* of AI-related scientific articles. This method involved keyword extraction and seeding, followed by machine learning-based optimization and clustering to identify and analyze trends within AI research.

Search strategy-based classification methods, while convenient, often have limitations in terms of recall, potentially missing relevant articles that do not fit neatly into predefined keyword categories. Traditional machine learning methods for classifying AI-related articles, such as using simple models like Random Forest or SciBERT, also face performance constraints. Their limitation lies in not fully utilizing certain important keywords or category names, which can lead to missing relevant content.

In contrast, our approach integrates keyword matching, decision tree based category classification, and the advanced capabilities of SciBERT. This combination allows for a more nuanced and comprehensive classification of AI-related articles, enhancing both accuracy and efficiency. By leveraging the strengths of each method, this integrated approach aims to provide a more robust and effective solution for classifying AI-related scholarly works.

## Decision tree

Decision trees are a fundamental machine learning technique used for classification and regression tasks. They operate by breaking down a dataset into smaller subsets while simultaneously developing an associated decision tree incrementally. The final result is a tree with decision nodes and leaf nodes, where each internal node represents a "test" on an attribute, each branch represents the outcome of the test, and each leaf node represents a class label (*Myles et al., 2004*).

The primary formula for decision tree algorithms involves the concept of information gain, which is derived from entropy. Entropy, a measure of disorder or uncertainty, is calculated using the formula:

$$\text{Entropy}(S) = -\sum_{i=1}^{n} p_i \log_2 p_i \tag{1}$$

where $p_i$ is the proportion of the number of elements in class $i$ to the number of elements in set $S$. Information gain, then, is defined as the difference in entropy before and after a dataset is split on an attribute. It is given by:

$$\text{Information Gain}(S, A) = \text{Entropy}(S) - \sum_{v \in \text{Values}(A)} \frac{|S_v|}{|S|} \text{Entropy}(S_v) \tag{2}$$

where $A$ is the attribute and $S_v$ is the subset of $S$ for which attribute $A$ has value $v$.

## SciBERT

BERT (Bidirectional Encoder Representations from Transformers) works on the principle of the Transformer, an attention mechanism that learns contextual relations between words in a text (*Devlin et al., 2018*). Unlike traditional models that read text sequentially (left-to-right or right-to-left), BERT reads the entire sequence of words at once. BERT is pre-trained on a large *corpus* of text and then fine-tuned for specific tasks. The Transformer model uses an attention mechanism that weighs the influence of different words on each other:

$$\text{Attention}(Q, K, V) = \text{softmax}\left(\frac{QK^T}{\sqrt{d_k}}\right)V \tag{3}$$

Here, $Q$, $K$, $V$ are queries, keys, and values respectively, and $d_k$ is the dimension of the keys.

BERT uses multiple layers of the Transformer, and each layer outputs transformed representations of the input text.

SciBERT adapts the BERT model specifically for scientific texts. It is pre-trained on a large *corpus* of scientific literature, encompassing a wide range of domains (*Beltagy, Lo & Cohan, 2019*). This specialization allows SciBERT to better understand and process the language used in academic and technical documents. It has been proven effective in tasks like classification (*Piao, 2021*), entity recognition (*Albared et al., 2019*), and relationship extraction (*Gangwar et al., 2021*) in scientific texts. As is mentioned before, it has been utilized in the classification of AI-related scientific literatures and patents (*Dunham, Melot & Murdick, 2020*; *Miric, Jia & Huang, 2023*). SciBERT's ability to grasp complex scientific concepts makes it particularly useful in classifying and analyzing scholarly articles, including those related to AI research.

## Support vector machine

SVM is a supervised machine learning algorithm used for both classification and regression. SVM works well for both linear and non-linear problems. The basic idea of SVM is to find the best hyperplane that separates data points of different classes in the feature space. In SVM, data points are plotted in a space where each feature is a dimension. The algorithm then identifies the optimal hyperplane that maximizes the margin between different classes. The data points that are closest to the hyperplane and influence its position and orientation are known as support vectors (*Jakkula, 2006*). The mathematical formulation of SVM involves finding the hyperplane that solves the following optimization problem:

$$\min_{w,b} 1/2 \parallel w \parallel^2 \tag{4}$$

Subject to: $y_i(\mathbf{w} \cdot \mathbf{x}_i + b) \geq 1$, for all $i$

Here, $w$ is the weight vector, $b$ is the bias, $x_i$ are the training examples, and $y_i$ are the class labels.

For non-linear problems, SVM uses kernel functions to transform the input space into a higher-dimensional space where a linear separation is possible. Common kernels include polynomial, radial basis function (RBF), and sigmoid.

SVM is particularly popular in text classification due to its effectiveness in high-dimensional spaces and its ability to handle overfitting, especially in cases where the number of features exceeds the number of samples (*Jakkula, 2006*).

## DATA COLLECTION

The volume of scientific literature available can make it difficult to identify relevant articles related to artificial intelligence, since researchers typically do not have full access to the entire WoS database. Therefore, we need to propose a search method to retrieve a portion of the WoS database for further analysis. To minimize the number of AI-related articles excluded from our study, it is necessary to ensure our search strategy includes as many AI-related articles as possible.

To begin with, we used the search term 'TS="Artificial Intelligence"' to retrieve all the articles related to AI from the Web of Science database. The publication time range was set from January 1, 2013 to December 31, 2022. The same time range was used for all of our future searches to ensure that we could analyze the AI-related articles from the past 10 years. A total of 95,835 results from Web of Science Core Collection were returned. We then conducted a high-frequency keyword analysis to identify the most common and relevant keywords related to AI. The details of the analysis are as follows: we first downloaded the Full records of all the articles returned above, and extracted "Author Keywords" and "Keywords Plus" fields from the records. Those keywords were ranked according to their total times of appearance. We then manually reviewed the keywords which appeared 200 times or more, and discarded those unrelated to AI. When necessary, we consulted wikipedia and WoS search results to clarify the meaning of a certain keyword. Since the aim of this step is to ensure a high recall rate for further classification, we retained a keyword whenever we were not sure if it would bring us more AI-related articles. As a result, a total of 196 keywords were retained.

To further ensure that we were not leaving any important search terms out, we also reviewed the search strategy used in previous studies. As is mentioned above, *Liu, Shapira & Yue (2021)* proposed a comprehensive search strategy for AI-related articles with widely used results. This consists of a core lexical query, two expanded lexical queries, and the WoS category "Artificial Intelligence". We also reviewed all WoS citation topics and determined 10 topics that belong to Artificial intelligence, including Natural Language Processing, Face Recognition, Defect Detection, Reinforcement Learning, Video Summarization, Action Recognition, Object Tracking, Deep Learning, Artificial Intelligence & Machine Learning and Visual Servoing. Our selected search terms were combined with the search terms proposed in *Liu, Shapira & Yue (2021)*, The final search strategy is shown in Table 1. This combined approach allowed us to retrieve a large number of AI-related articles that were relevant to our research objectives.

To make sure our results are comparable with previous results, we used our search strategy in the WoS Science Citation Index Expanded (SCI-Expanded) and Social Sciences

**Table 1 Preliminary search approach for artificial intelligence.** Data were retrieved from WoS database with the following search strategies for further analysis.

| Author | Search strategy | Search terms |
|---|---|---|
| *Liu, Shapira & Yue (2021)* | Core lexical query | TS=("Artificial Intelligen*" or "Neural Net*" or "Machine* Learning" or "Expert System$" or "Natural Language Processing" or "Deep Learning" or "Reinforcement Learning" or "Learning Algorithm$" or "Supervised Learning" or "Intelligent Agent") |
| | Expanded lexical query 1 | TS=(("Backpropagation Learning" or "Back-propagation Learning" or "Bp Learning") or ("Backpropagation Algorithm*" or "Back-propagation Algorithm*") or "Long Short-term Memory" or ((Pcnn$ not Pcnnt) or "Pulse Coupled Neural Net*") or "Perceptron$" or ("Neuro-evolution" or Neuroevolution) or "Liquid State Machine*" or "Deep Belief Net*" or ("Radial Basis Function Net*" or Rbfnn* or "Rbf Net*") or "Deep Net*" or Autoencoder* or "Committee Machine*" or "Training Algorithm$" or ("Backpropagation Net*" or "Back-propagation Net*" or "Bp Network*") or "Q learning" or "Convolution* Net*" or "Actor-critic Algorithm$" or ("Feedforward Net*" or "Feed-Forward Net*") or "Hopfeld Net*" or Neocognitron* or Xgboost* or "Boltzmann Machine*" or "Activation Function$" or ("Neurodynamic Programming" or "Neuro dynamic Programming") or "Learning Model*" or (Neuro computing or "Neuro-Computing") or "Temporal Difference Learning" or "Echo State* Net*") |
| | Expanded lexical query 2 | TS=("Transfer Learning" or "Gradient Boosting" or "Adversarial Learning" or "Feature Learning" or "Generative Adversarial Net*" or "Representation Learning" or ("Multiagent Learning" or "Multi-agent Learning") or "Reservoir Computing" or "Co-training" or ("Pac Learning" or "Probabl* Approximate* Correct Learning") or "Extreme Learning Machine*" or "Ensemble Learning" or "Machine* Intelligen*" or ("Neuro fuzzy" or "Neurofuzzy") or "Lazy Learning" or ("Multi* instance Learning" or "Multi-instance Learning") or ("Multi* task Learning" or "Multitask Learning") or "Computation* Intelligen*" or "Neural Model*" or ("Multi* label Learning" or "Multilabel Learning") or "Similarity Learning" or "Statistical Relation* Learning" or "Support* Vector* Regression" or "Manifold Regularization" or "Decision Forest*" or "Generalization Error*" or "Transductive Learning" or (Neurorobotic* or "Neuro-robotic*") or "Inductive Logic Programming" or "Natural Language Understanding" or (Ada-boost* or "Adaptive Boosting") or "Incremental Learning" or "Random Forest*" or "Metric Learning" or "Neural Gas" or "Grammatical Inference" or "Support* Vector* Machine*" or ("Multi* label Classification" or "Multilabel Classification") or "Conditional Random Field*" or ("Multi* class Classification" or "Multiclass Classification") or "Mixture Of Expert*" or "Concept* Drift" or "Genetic Programming" or "String Kernel*" or |
| | | ("Learning To Rank*" or "Machine-learned Ranking") or "Boosting Algorithm$" or "Robot* Learning" or "Relevance Vector* Machine*" or Connectionis* or ("Multi* Kernel$ Learning" or "Multikernel$ Learning") or "Graph Learning" or "Naive bayes* Classif*" or "Rule-based System$" or "Classification Algorithm*" or "Graph* Kernel*" or "Rule* induction" or "Manifold Learning" or "Label Propagation" or "Hypergraph* Learning" or "One class Classif*" or "Intelligent Algorithm*") |
| | WoS category | WC=("Artificial Intelligence") |
| This study | Lexical query | See Appendix. |
| | Citation topics | 4.48.672 Natural Language Processing |
| | | 4.48.672 Natural Language Processing |
| | | 4.17.118 Face Recognition |
| | | 4.17.1950 Defect Detection |
| | | 4.116.862 Reinforcement Learning |
| | | 4.17.1802 Video Summarization |
| | | 4.17.630 Action Recognition |
| | | 4.17.953 Object Tracking |
| | | 4.17.128 Deep Learning |
| | | 4.61 Artificial Intelligence & Machine Learning |
| | | 4.116.2066 Visual Servoing |
| | WoS category | WC=("Artificial Intelligence") |

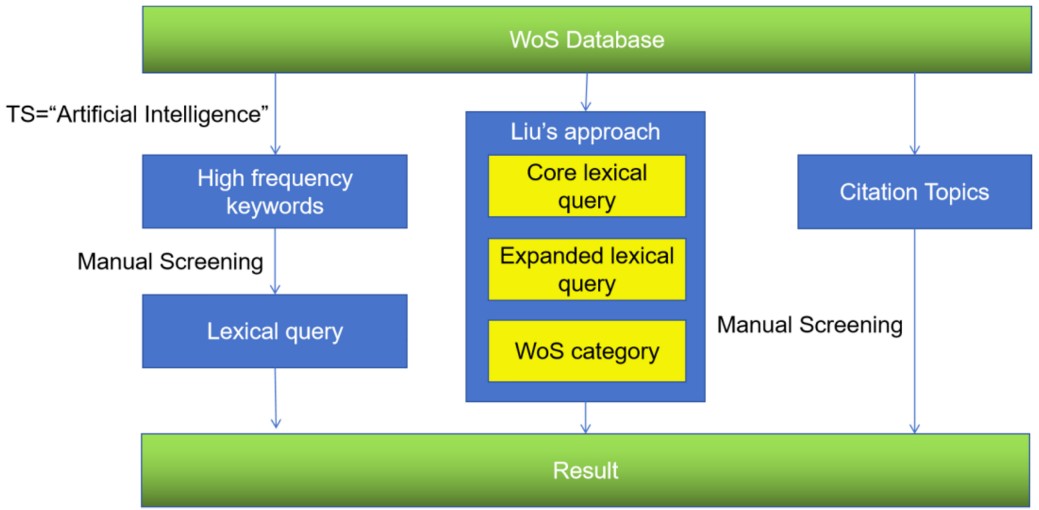

**Figure 1 Overview of the data collection process, including high-frequency keywords analysis, the search strategy of *Liu, Shapira & Yue (2021)*, and citation topics.**

Citation Index (SSCI) databases, and set the same time limit January 1, 2013 to December 31, 2022. 662,844 articles were retrieved with the approach of *Liu, Shapira & Yue (2021)*. When combined with our proposed approach, there are a total of 2,490,817 articles for further screening and analysis. The results indicate that most of the AI-related articles from WoS cannot be captured by a simple search of TS="Artificial Intelligence".

The process of data collection is shown in Fig. 1.

We then downloaded the *corpus* (referred to as the initial *corpus*) *via* the search strategy described in Table 1. In order to obtain the training data, we sampled 4,000 articles from the initial *corpus* and manually labelled them as "AI-related" and "Other". The definition of AI used in this study is a technology in which some form of statistical learning is employed, and perceives its environment to predict an action that maximizes its chance of achieving its goals (*Cockburn, Henderson & Stern, 2018*). This definition was selected because the article provides very detailed explanation and large number of examples of AI-related technology in its full text. We enrolled two computer science students as coders to independently label the articles. The coders were instructed to follow the aforementioned definition and read through the title and abstract to determine whether an article is AI-related. The authors reviewed their classification results and discussed with the coders to align their understandings of AI and reduce their differences. Finally, when disagreements between the coders cannot be resolved, an AI expert from Zhejiang University was consulted to determine the final result.

## MODEL DEVELOPMENT

Our proposed model integrates multiple approaches to effectively classify AI-related research articles. It uses keyword matching to scan the keywords of each article, a decision tree to classify articles based on their category, and SciBERT to analyze the title and

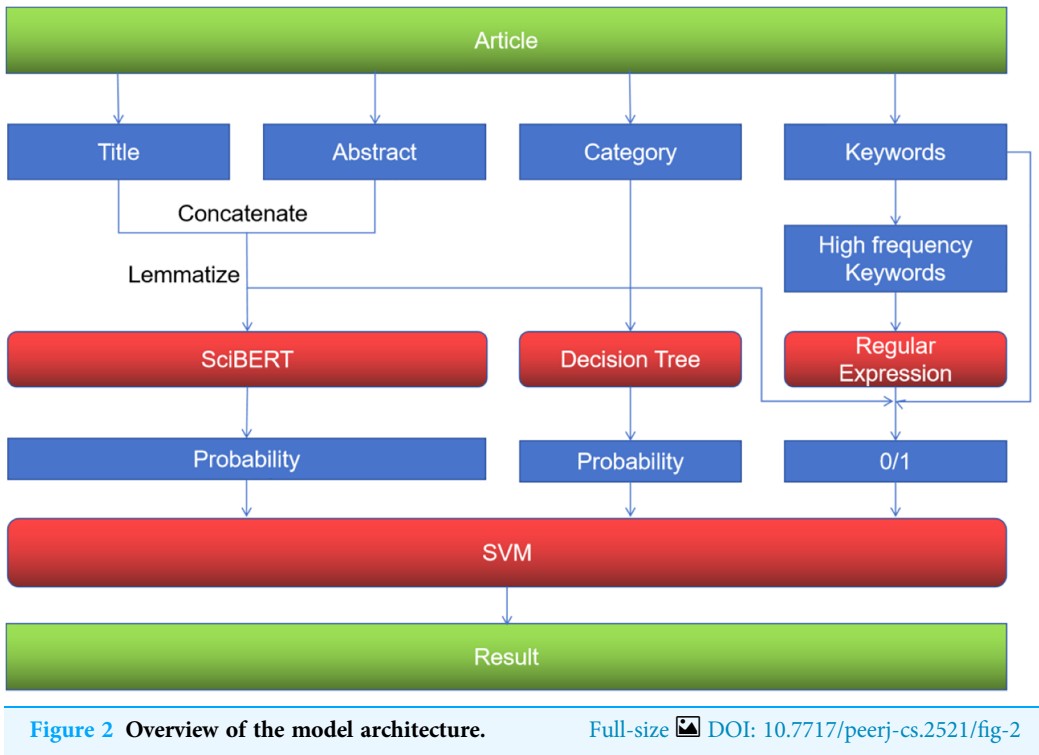

**Figure 2  Overview of the model architecture.** 

abstract of each article. These components are then combined using a SVM to provide a final classification.

Initially, we experimented with replacing the decision tree with more complex ensemble methods such as Random Forest and XGBoost. However, in our case, the category data was relatively straightforward, since the category "artificial intelligence" is already present in the WoS categories. Using Random Forest and XGBoost led to overfitting because these models tend to find complex patterns in training data and ignore the importance of WoS category "artificial intelligence".

Similarly, manually screening the keywords proved to be more effective than using other models such as naive Bayes or latent Dirichlet allocation (LDA). The main limitation of naive Bayes in this context is that it is not direct enough. Instead of simply identifying relevant AI-related keywords like "deep learning" or "natural language processing," naive Bayes calculates the probability of an article being AI-related based on word frequencies, which results in lower performance than keyword matching based on a set of manually curated keywords. On the other hand, LDA, which is a topic modeling approach, groups articles into several latent topics based on word distributions. However, none of these topics align perfectly with the concept of Artificial Intelligence as a research field.

To merge the results of the individual components—keyword matching, decision tree, and SciBERT—we selected SVM as the final classifier. A simpler approach, such as logistic regression or majority voting, would lack the flexibility needed to combine the results from such diverse components effectively. Alternatively, more complex models like neural networks could potentially overfit, given the small number of features.

Overview of the model architecture is shown in Fig. 2.

## SciBERT

In our approach to classifying scientific articles, we leveraged SciBERT, a state-of-the-art pretrained language model specifically designed for scientific text. For each article in our manually labeled dataset, which served as the ground truth, we concatenated and lemmatized its title and abstract to form a single input text sequence. We then fine-tuned sciBERT using this concatenated text, with the following hyperparameters: number of epochs: three, training batch size: eight, evaluation batch size: eight, warmup steps: 500, weight decay: 0.01, learning rate: 5e-, optimizer: AdamW. A five-fold cross-validation was used in the fine-tuning process. In each fold, four sets were used for training and the remaining set for validation. The output of the SciBERT fine-tuning for our task is a probability score between 0 and 1 for each article, representing the likelihood that the article is AI-related. This is achieved by selecting the sigmoid activation function as the final layer of the sciBERT model.

Let $T$ and $A$ be the title and abstract of an article respectively. The probability score $P$ indicating whether the article is AI-related is obtained by:

$$P = \sigma(W \cdot \text{SciBERT}(\text{Lemmatize}(T + A)) + b) \tag{5}$$

$$\sigma(x) = \frac{1}{1 + e^{-x}} \tag{6}$$

Here, $\sigma$ is the sigmoid function, $W$ is the weight matrix, and is the bias term.

The fine-tuned SciBERT model constitutes the foundation of our article classification system. Building upon this base, we incorporated additional components aimed at enhancing the model's predictive accuracy. These components leveraged other article attributes and were integrated through a ensemble method.

## Decision tree

In parallel to the fine-tuning of SciBERT, we sought to exploit the categorical metadata associated with scientific articles. The Web of Science (WoS) categories assigned to each article offer a high-level view of the content domain. To make use of this categorical information, we employed a decision tree classifier.

This classifier was trained to predict the likelihood of an article being related to artificial intelligence based on its WoS categories. Decision trees are particularly advantageous for their interpretability and ease of use. To validate the effectiveness of this approach, the 4,000 labeled articles from our dataset were subjected to a five-fold cross-validation scheme, akin to the method applied during the SciBERT fine-tuning process. A maximum depth of two was used to limit the complexity of the model. Other hyperparameters were kept at their default values.

At each node $n$ of the tree, the algorithm selects the best feature $f$ and a threshold $\theta$ that maximizes information gain. This is represented as:

$$\text{Split}(n) = \text{argmax}_{f,\theta} \text{ InformationGain}(D, f, \theta) \tag{7}$$

where Information Gain is calculated as the difference in entropy (measure of uncertainty or randomness) before and after the split. Once the tree is built, the probability of an article

**Table 2  AI-related keywords used in the keyword matching section.**

**Regular expression**

(Artificial Intelligen.*|Machine Learning|Deep Learning|Genetic Algorithm|Support Vector Machine|
Image Segmentation|Particle Swarm|Reinforcement Learning|Random Forest|Computer Vision|Transfer
Learning|Natural Language Processing|Supervised Learning|Semantic Segmentation|Generative Adversarial
Network|Sentiment Analysis|Multi-Agent System|MultiAgent System|Ensemble Learning|Extreme Learning|
Recommender System|Image Retrieval|Decision Tree|image Fusion|Long Short-Term Memory|Evolutionary
Algorithm |Ant Colony Optimization|Convolutional Neural Network|Artificial Neural Network|Deep Neural
Network|Recurrent Neural Network|BP Neural Network|Graph Neural Network|\bSVM\b|\bNLP\b|
\bLSTM\b|\bCNN\b|\bDNN\b|\bRNN\b|\bGNN\b)

being AI-related is estimated based on the proportion of AI-related articles in the leaf node where the article falls.

## Keyword matching

To further refine our classification model, we conducted an in-depth analysis of the high-frequency keywords within the initial *corpus*. This involved a manual review of the high-frequency keywords in the initial *corpus* to identify AI-related ones. Our goal was to establish a list of AI-related keywords that could serve as strong indicators for classifying articles as relevant to the field of AI.

For each keyword identified in this process, we randomly selected 20 articles containing the keyword from the initial *corpus*. This sample was scrutinized not only by the authors but also by external AI experts to ensure that the presence of the keyword was indeed indicative of the article's relevance to artificial intelligence. If all of the 20 articles were considered AI-related, this keyword would be added to our keyword list. We were able to curate a list of keywords with a high degree of confidence in their relevance to artificial intelligence research. The keywords were summarized into the regular expression in Table 2. We then performed regular expression matching on title, abstract and keywords of articles to determine whether it contains any AI-related terms, and the matching results were included into our ensemble model.

## SVM

Building upon the individual strengths of SciBERT, the decision tree, and keyword matching, we integrate these components using a SVM. The SVM acts as a meta-classifier, taking as input the probability outputs from SciBERT and decision tree, and the binary results from the regular expression keyword matching. The SVM is trained to find an optimal boundary that separates AI-related articles from others. The SVM used a linear kernel, and the regularization parameter C was set to its default value of 1.0. For each article, the input features include the probability score from SciBERT, indicating the probability of the article being AI-related; the output of the decision tree, which predicts the probability based on the article's categorical data; and a binary feature signifying the presence or absence of AI-related terms as determined by our regular expression analysis.

Let $T_x$, $A_x$, $K_x$, $C_x$ and represent the title, abstract, keywords and WoS categories of article $x$ respectively, the SVM function incorporating these specific inputs would be represented as:

$$y = \text{SVM}(P_{\text{sciBERT}}(T_x + A_x), P_{DT}(C_x), I_{\text{regex}}(T_x + A_x, K))\tag{8}$$

And $I_{regex}$ is the binary output from the regular expression matching.

# RESULTS

## Performance evaluation

At the beginning of this section, we examine the overlap between the different approaches used to identify AI-related articles. As is mentioned above, approaches in retrieving AI-related articles range from simple search term "Artificial Intelligence", WoS category "Computer Science, Artificial Intelligence", complicated search terms, to employing deep learning models. Three strategies were selected for further comparison: keyword based search strategy by *Liu, Shapira & Yue (2021)* (referred to as Liu's approach), WoS Category="Artificial Intelligence" Approach (referred to as WoS Category approach), and our Ensemble model (referred to as the EnsembleAI model). We selected Liu's search strategy because it achieves the highest performance among existing search strategies (*Liu, Shapira & Yue, 2021*). The simplest approach of using the query TS=("artificial intelligence") to search the WoS database yielded a very low recall, therefore it is omitted in the following analysis. Figure 3 illustrates the coverage and intersection of the three approaches.

The WoS category approach, which contains 192,067 articles, is included in both Liu's approach and EnsembleAI. Liu's approach is almost included in EnsembleAI, with the exception of 24,606 articles. Our EnsembleAI approach contains the biggest number of articles among the three approaches, with 402,962 unique articles. The significant overlap between the approaches indicate that we share a common understanding of the AI research landscape. However, the disparities in the number of articles identified by Liu's approach and EnsembleAI suggest that there may be unique aspects captured by our methods.

Next, we evaluated the performance of various methods to identify AI-related articles. Previously manually annotated dataset of 4,000 articles were directly used to evaluate the performance of WoS category and Liu's approach. For EnsembleAI, 5-fold cross validation was employed during training and performance evaluation was done by applying the corresponding test set in each fold and computing the average performance.

The performance of each method is evaluated in terms of precision and recall. Precision measures the fraction of true AI-related articles among those identified by the method, while recall measures the fraction of true AI-related articles identified by the method out of all the AI-related articles in the sample.

The following table summarizes the results:

Table 3 presents the performance metrics of three distinct approaches for classifying articles as related to artificial intelligence. Precision measures the proportion of true positive results among all positive cases identified by the classifier, while recall (or

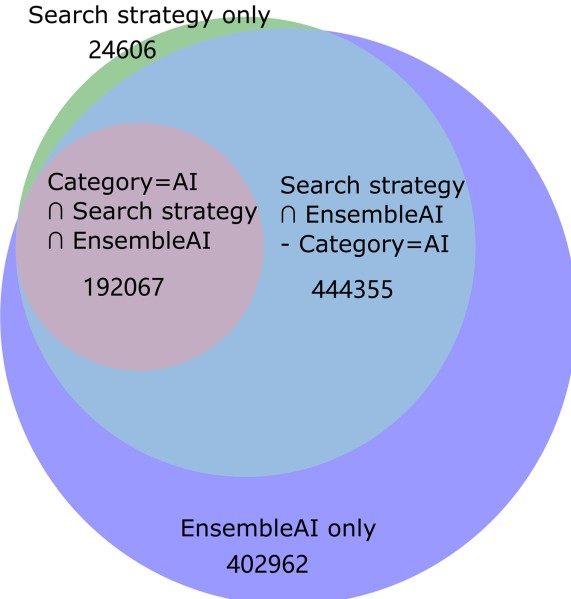

**Figure 3** The number of articles extracted from the WoS database using different AI classification approaches, including Liu, Shapira & Yue's (2021) search strategy-based approach, WoS category= 'Artificial Intelligence' approach, the ensembleAI model proposed in this article.

**Table 3 Performance of different classification approaches.**

| Name | Precision | Recall | F1-Score |
|------|-----------|--------|----------|
| WoS category | 0.934 | 0.325 | 0.482 |
| Liu's approach | 0.980 | 0.669 | 0.795 |
| EnsembleAI | 0.915 | 0.973 | 0.943 |

sensitivity) measures the proportion of true positive results among all actual positive cases. The F1-score is the harmonic mean of precision and recall, providing a single metric that balances both.

The WoS category-based approach shows precision of 0.934, since not all texts in this category were classified as AI related by our experts. For example, articles dealing with human-machine interfaces, chemometrics and bilevel optimization were deemed non-AI by our experts, and we excluded them from the AI-related articles. Its recall is quite low (0.325), suggesting it fails to identify a large number of AI-related articles. Consequently, its F1-score is relatively low (0.482), reflecting the imbalance between precision and recall.

Liu's approach significantly improves upon the WoS category, particularly in recall (0.669), while maintaining excellent precision (0.980). This balance results in a much higher F1-score (0.795), suggesting a more effective model overall.

The EnsembleAI shows a slight decrease in precision (0.915) compared to Liu's approach, but has a superior recall (0.973). This high recall indicates that EnsembleAI is particularly adept at identifying most AI-related articles. Despite the slight trade-off in

**Table 4 Comparison of the performance of models with different fields as input.** Decision tree, keywords, SVM were individually excluded from the model, and title and abstract were individually excluded from the sciBERT.

|  | EnsembleAI | EnsembleAI-decision tree | EnsembleAI-keywords | EnsembleAI-SVM | EnsembleAI-title | EnsembleAI-abstract |
|---|---|---|---|---|---|---|
| Precision | 0.915 | 0.921 | 0.915 | 0.902 | 0.905 | 0.930 |
| Recall | 0.973 | 0.914 | 0.959 | 0.978 | 0.970 | 0.936 |
| F1-score | 0.943 | 0.918 | 0.937 | 0.939 | 0.936 | 0.933 |

precision, EnsembleAI achieves the highest F1-score (0.943) among the three methods, indicating a robust performance and a well-balanced approach between precision and recall.

To account for AI-related articles not included in our initial search strategy, we randomly downloaded 2,500 articles from the WoS and subjected them to our ensembleAI model. The random sampling is implemented by randomly selecting a day from last year, downloading all articles published on that day, and then randomly select 2,500 articles from the result. An article is considered AI-related if it is classified as relevant by the ensembleAI model. Out of the 110 relevant articles in the 2,500 random samples, 105 of them are included in the final *corpus*. The results indicates that our search strategy successfully captures 95% of all the AI-related articles in the entire WoS *corpus*.

## Ablation study

The ablation study will methodically examine the EnsembleAI model by selectively removing or isolating specific components, such as the decision tree, sciBERT, keyword matching, and the SVM integrator. This process aims to reveal the impact and value of each component within the ensemble framework. Through this study, we intend to quantify the contribution of each part of the EnsembleAI model to its precision, recall, and F1-score.

Overall, the ablation study highlights the synergistic effect of the combined components in the EnsembleAI model. Each component contributes uniquely to the model's performance, with the decision tree and abstract elements being particularly crucial for maintaining high recall, and the SVM integrator more significantly affecting precision. The balanced performance of the original EnsembleAI model underscores the importance of integrating multiple approaches for effective identification of AI-related articles.

The data from the ablation study of the EnsembleAI model provides insightful revelations about the contributions of its individual components to the overall performance in identifying AI-related articles. Table 4 shows the performance data, followed by an analysis of each variant:

EnsembleAI (Original Model): With a precision of 0.915, recall of 0.973, and an F1-score of 0.943, the original EnsembleAI model exhibits a well-balanced performance.

EnsembleAI-Decision Tree: Removing the decision tree component slightly increases precision to 0.921 but notably reduces recall to 0.914. The F1-score drops to 0.918.

**Table 5 Comparison of the performance of models with different BERT-based models.**

|  | EnsembleAI | sciBERT | BERT-base | RoBERTa | SciBERT (all) |
|---|---|---|---|---|---|
| Precision | 0.915 | 0.928 | 0.919 | 0.911 | 0.935 |
| Recall | 0.973 | 0.896 | 0.877 | 0.879 | 0.904 |
| F1-score | 0.943 | 0.911 | 0.898 | 0.895 | 0.919 |

EnsembleAI-Keywords: The removal of keyword matching maintains the same precision (0.915) but slightly decreases recall to 0.959. The F1-score also sees a slight decrease to 0.937.

EnsembleAI-SVM: Excluding the SVM integrator decreases precision to 0.902 and slightly increases recall to 0.978. The F1-score is relatively stable at 0.939. Removing SVM is done by retaining the results from sciBERT and decision tree with a probability of 0.5 and more, and computing the union of positive results from sciBERT, decision tree and keywords matching. This method outperformed majority voting and soft voting, therefore it was selected as the way to replace the SVM integrator.

EnsembleAI-title: Removing the title from the SciBERT input lowers precision slightly to 0.905 and recall to 0.970, with an F1-score of 0.936.

EnsembleAI-abstract: Excluding the abstract from SciBERT input results in the highest precision (0.930) but lowers recall to 0.936, leading to an F1-score of 0.933.

Furthermore, we explored the effectiveness of using only the SciBERT component, as well as the impact of replacing SciBERT with other BERT variants when classifying AI-related articles.

BERT-base, developed by Google, was one of the first models to use Transformer architecture, significantly changing the landscape of text-based analysis by providing a deep, bidirectional understanding of context within language. It is pre-trained on a vast *corpus* of text and designed to be fine-tuned for a variety of Natural language processing (NLP) tasks (*Devlin et al., 2018*).

RoBERTa, which stands for Robustly Optimized BERT Pretraining Approach, is a model developed by Facebook AI that builds upon BERT's architecture. It has been optimized with more data, larger batch sizes, and longer training times, leading to improved performance over BERT-base in many benchmarks (*Liu et al., 2019*).

BERT-base and RoBERTa were selected as alternative models and we conducted a series of additional experiments of feeding different combinations of textual data into these models, including title + abstract, and concatenated fields of title + abstract + WoS categories + keywords. The results are shown in Table 5.

EnsembleAI: This is the original EnsembleAI model with a precision of 0.915, recall of 0.973, and F1-score of 0.943. It serves as the baseline for comparison with other model configurations.

sciBERT: By only utilizing sciBERT with the titles and abstracts, we see a slight increase in precision to 0.928 but a notable drop in recall to 0.896. The F1-score of 0.911 indicates

that while sciBERT alone is effective, the additional components in the EnsembleAI model contribute to overall performance.

BERT-base: Replacing sciBERT with BERT-base leads to a precision of 0.919 and a further decrease in recall to 0.877. The F1-score drops to 0.898, suggesting that BERT-base is less effective at identifying AI-related articles than the tailored sciBERT in the context of this task.

RoBERTa: With RoBERTa, there is a slight decrease in both precision and recall compared to BERT-base, resulting in an F1-score of 0.895.

sciBERT (all): This configuration, which feeds a concatenation of title, abstract, WoS category, and keywords directly into sciBERT, achieves a precision of 0.935 and a recall of 0.904. The F1-score improves to 0.919, indicating that incorporating category and keywords information directly into sciBERT's input can enhance performance, though it does not reach the efficacy of the full EnsembleAI model.

These results affirm that while sciBERT is a powerful tool for classifying AI-related articles, the combination of decision trees, keyword matching, and an SVM integrator in the EnsembleAI model leads to superior recall and the best overall performance as measured by the F1-score. Each component of the EnsembleAI model contributes uniquely to its high level of accuracy and ability to comprehensively identify relevant articles, illustrating the benefits of a multifaceted approach to article classification in rapidly evolving research fields like AI.

## DISCUSSION

In this study, we proposed a combined approach that integrates a decision tree, SciBERT, and keyword matching across various fields, followed by the use of a SVM to amalgamate the results. We then identified scholarly articles belonging to the field of artificial intelligence. The performance of our approach was evaluated with manually labeled data from the initial *corpus*. The results indicate that we achieved significantly increased recall and similar precision, when compared with strategies from previous research.

We noted that previous research has employed machine learning on classifying AI research articles (*Dunham, Melot & Murdick, 2020*; *Siebert et al., 2018*). *Siebert et al. (2018)* achieved a self-reported accuracy of 85%, but when applied to arXiv data, *Dunham, Melot & Murdick (2020)* estimated that *Siebert et al.*'s *(2018)* approach has a precision of 74% and recall of 49%. *Dunham, Melot & Murdick*'s *(2020)* approach had a precision of 83% and recall of 85%, but their estimation is based on existing categories. Thus, the fact that only a portion of AI-related publications belong to the 'Artificial Intelligence' category would make the actual recall significantly lower than the reported values in *Dunham, Melot & Murdick*'s *(2020)* approach. In conclusion, our approach has the best performance among the approaches employing machine learning to classify AI-related research articles.

While our study provides valuable insights, there are limitations to our approach. As is mentioned earlier, the concept of artificial intelligence itself lacks a clear and widely accepted definition (*Wang, 2019*). Previous definitions vary from "The science of making machines do things that would require intelligence if done by men" (*Minsky, 1965*) to "the endowment of machines with human-like capabilities through simulating human

consciousness and thinking processes using advanced algorithms or models"
(*Jakhar & Kaur, 2020*). In this study, we selected the definition with the most detailed
explanation and the highest number of specific examples to our knowledge. We tried using
different definitions, and the performance evaluation results can vary. We labeled a small
data set with different definitions and the Cohen's Kappa appeared to be around 0.9,
indicating a very high level of agreement. Therefore, the differences were small and did not
affect our conclusions. In addition, our approach relies partially on the existing WoS
category "Artificial Intelligence". We would need to modify our model if we try to apply it
to other emerging fields where corresponding WoS categories do not exist.

In addition to title, abstract, keywords and WoS categories, we also attempted to include
other fields such as journal name or author name into the model. But results showed that
they were weak indicators of AI relevance, and including them into the SVM would lower
the performance. They were therefore excluded in this study. But with more advanced
modelling and feature engineering techniques, they may prove useful in the classification
of research articles in future research.

In light of the growing importance of AI, future work can extend our approach to the
analysis of patent data in AI and text-mining of websites. This proposition is supported by
the findings of previous studies. Researchers have conducted a landscape analysis of AI
innovation dynamics and technology evolution using a new AI patent search strategy,
incorporating patent analyses, network analyses, and source path link count algorithms
(*Liu et al., 2021*). Our ensemble approach can be applied to further enhance the
understanding of AI patenting trends and cross-organization knowledge flows. Another
study (*Arora et al., 2020*) employed topic modeling, a text-mining approach, on archived
website data to investigate sales growth for green goods enterprises. This study
demonstrated the potential of website data to gauge internal capabilities and market
responsiveness. By utilizing the natural language processing ability of the ensemble models
to enhance text-mining performance, future work can unlock new insights into the
strategic management of innovation and entrepreneurship. Furthermore, our approach
can be directly applied to other fields of research, such as nanotechnology (*Wang et al.,
2019*), synthetic biology (*Shapira, Kwon & Youtie, 2017*) and cancer research (*Yeshawant
& Ravi, 2016*).

## CONCLUSION

In conclusion, this study presents an ensemble approach to address the challenges of text
identification in the rapidly evolving field of AI. Our approach demonstrates a high
precision of 92% and successfully captures about 97% of AI-related articles in the Web of
Science (WoS) *corpus*. By comparing our approach with existing search-term based
methods, we conclude that our approach yielded a similar precision and significantly
increased recall, and resulted in a 0.15 increase in F1-score.

This study highlights the potential of our comprehensive approach in facilitating
accurate text classification in emerging research fields like AI. By incorporating ensemble
models into the text-mining process of different data sources, various stakeholders, such as
researchers, policymakers, and industry practitioners, can be enabled to leverage the power

of deep learning models for future research, policy decisions, and technological advancements.

## ACKNOWLEDGEMENTS

We thank professor Philip Shapira, Dr. James Dunham and Lizhou Fan for their discussion.

### Funding

The authors received no funding for this work.

### Competing Interests

The authors declare that they have no competing interests.

### Author Contributions

- Min Lu conceived and designed the experiments, analyzed the data, prepared figures and/or tables, authored or reviewed drafts of the article, and approved the final draft.
- Lie Tang conceived and designed the experiments, performed the experiments, analyzed the data, performed the computation work, prepared figures and/or tables, authored or reviewed drafts of the article, and approved the final draft.
- Xianke Zhou performed the experiments, authored or reviewed drafts of the article, and approved the final draft.

### Data Availability

The code is available at GitHub and Zenodo:

- https://github.com/tanglie1993/WOS_AI_ensemble

- tanglie. (2024). tanglie1993/WOS_AI_ensemble: 1.0.0 (publish). Zenodo. https://doi.org/10.5281/zenodo.13822157

The full data is available at Zenodo:

Tang, L. (2024). Full data [Data set]. Zenodo. https://doi.org/10.5281/zenodo.14195932.

### Supplemental Information

Supplemental information for this article can be found online at http://dx.doi.org/10.7717/peerj-cs.2521#supplemental-information.

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
