# Peer review of "An ensemble approach for research article classification: a case study in artificial intelligence"

_PeerJ Computer Science, doi:10.7717/peerj-cs.2521_

## Round 0.1 · original submission · Major Revisions

An ensemble approach for research article classification: a case study in artificial intelligence

Summary:
The authors address the task of classifying documents as AI or non-AI related. They proposed an ensemble approach that combines a decision tree algorithm to categorize articles based on a hierarchical set of Web of science categories, SciBERT, a BERT language model specifically trained on scientific literature, to understand content of articles, and expert-crafted keyword matching. They used an SVM model to merge the results from these three methods for the final classification. They evaluated their ensemble approach on a collection of 4,000 manually labeled research articles and showed high performance (precision 0.92) which improves the state of the art by 0.15.

Comments:
The authors proposed a method for scientific document classification, a well-studied problem (https://scholar.google.com/scholar?hl=en&as_sdt=0%2C47&q=%22document+classification%22&btnG=), by combining three well-known NLP/machine or deep-learning learning techniques (decision tree, SciBERT, keyword matching with regular expression). However, the the ensemble of these models presents a novel contribution in classifying research articles in specific domains.

The authors should address all of the issue raised in the reviewer comments.

Reviewer 1 ·

Basic reporting

This paper presents a deep learning-based ensemble approach for research article classification. This approach integrates a decision tree, SciBERT, and keyword matching, with the results from each model being amalgamated using a Support Vector Machine. The authors evaluated the effectiveness of their method through a case study in the field of Artificial Intelligence.

I am concerned that the clarity of the illustration in Figure 3 may not be sufficient for readers. What does each circle in the figure 3 represent? The figure needs to be improved by annotating each circle.

I suggest that the authors improve the figure and table captions to include the necessary details while remaining concise. For example, it would be better if the caption of Figure 1 were revised to: 'Overview of the data collection process, including high-frequency keywords analysis, Liu’s search strategy, and citation topics.' Additionally, the caption of Figure 3 should state what the numbers in the figure represent. For example: 'The number of articles extracted from the WoS database using different AI classification approaches, including Liu’s approach, WoS category approach, etc.'

The manuscript is clearly written in professional and unambiguous language. However, there are some typos and formatting issues. For instance, the opening quotation marks for all single and double quotations need to be corrected. In the introduction section, the sentence “Emerging research fields, characterized by complex boundaries and rapid evolution(WIPO, 2019). present unique challenges for accurate identification of research articles.” has is a full stop (.) in the middle of a sentence. I recommend that the authors address such typos and formatting issues.

Experimental design

Data collection process is clearly explained and depicted as a figure in the manuscript. The authors have employed multiple methods to search the articles related to AI. Data preprocessing is briefly discussed for the SciBERT model development. However, did the authors conduct any data preprocessing or cleaning for decision making and keyword matching models? I suggest the authors to discuss how they conducted data processing for all the models.

The authors have explained why the selected machine learning methods are effective for a text classification study, and they have also experimented with different BERT variants to justify their selection of the SciBERT model. However, did they experiment with other machine learning techniques before choosing decision tree and SVM methods? A detailed explanation of how they methodically selected decision tree and SVM techniques from other machine learning options would be helpful.

The authors have clearly explained the performance evaluations they have conducted. They have also described the performance metrics they have used to compare different approaches for classifying papers and why those metrics are effective.

Validity of the findings

This study has replicated the results from previous classification approaches, namely the WoS category and Liu’s approach, and compared them against the method introduced by the authors, Ensemble AI, in terms of precision, recall, and F1-score.

The authors have also conducted variety of experiments in this study.
1. Preliminary search approach for AI to show that the search term “Artificial Intelligence” is not sufficient.
2. Ablation study to methodically evaluate the performance of the ensemble approach and the impact of each component of the ensemble approach.
3. Experiment to explore the effectiveness of the selected model by comparing the performance of different BERT variants.

This study evaluated performance based on precision, recall, and F1-score. The model demonstrated strong performance, particularly in terms of the F1-score, compared to the other approaches in the evaluation.

The authors have discussed the limitations of their approach and how it can be extended in the future in the discussion section. They have also addressed the potential implications and applications of the research across various domains. However, I recommend that the authors include their future directions and plans for this study in the conclusion section.

Reviewer 2 ·

Basic reporting

a) Poor Abstract:
-- Examples of previous methods could help understand the difference between the proposed and significant contributions. For example, in line 12, the author claimed, "traditional methods, which rely on manually curated....". In line 14, the author says, "these limitations"; however, there is only one limitation. What are the other types of limitations?
-- The dataset collection is unclear. For example, in line 18, the authors manually labeled the dataset. What kind of dataset was used for this experimentation? The dataset compilation strategy is unclear.
-- The author claimed their proposed model helped improve "precision" compared to previous traditional methods. However, what about recall? Did the 'recall' improve since the authors claimed "poor recall in traditional method" in line 13?
b) The introduction section discusses many advancements in AI. However, it fails to show the "list of contributions." For example, I could not understand the dataset collection strategy from WoS until the Data Collection section. A short paragraph would help in this case.
c) The thesis statement in the introduction is unclear. I read through the method section to understand the author's goal.
d) I was still unclear why the authors manually labeled 4,000 articles. Also, what disciplines and domains did they choose to label 4,000 research articles? A dataset distribution table would help in this case. Again, I had to read through the 'data collection' section to understand from where these 4,000 articles were chosen.
e) The Citations at the end of the long paragraph in the literature review are confusing. For example, in line 112, the authors imply 'bibliographic indicators to detect research papers in emerging fields in another study.' There is no citation after that, except at the end of the whole paragraph. This is ambiguous referencing.

Experimental design

Data Collection:
a) In the dataset section, at line 217, what does the author mean by high-frequency keyword analysis? Is there an example, reference, etc.?
b) What ranking method have the authors used to rank keywords based on their time of appearance? Is that a simple tf-idf or any other deep-learning technique?
c) I understood that the author collected many research articles using Liu's, WoS, and Tang's approaches. However, I am unsure of the domain and disciplines of this research article. Dataset distribution is expected.
d) In line 221, the authors "manually reviewed the keywords that appeared 200 times or more". It is a very ambiguous strategy for keyword inspection. Why not 300, 400, or more?
e) Have you explored 'topic modeling' to identify the articles related to AI instead of manually checking the keywords?
f) An ontology-based approach could also be used to identify AI-related articles. Have you explored such an automatic approach?
g) In Figure 1, the author could use a robust approach, such as an unsupervised technique or relation extraction instead of manual screening.
h) The authors collected a large amount of data (~2M with Tang's approach and ~600K with Liu's approach, etc.). Only 4,000 articles were chosen. If I choose a different set of 4,000 articles, will it produce the same result? This is a concern of reproducibility.
i) What's the dataset distribution of these 4,000 articles? Are these from Social Science, Public Health, Computer Science, etc?
j) Poor strategy to cross-verify the annotation (line 249 -- line 254). The annotation could be evaluated by reporting the metric of IOU (intersection over union).

Model Development:
a) In line 257, the authors claimed SciBERT is a state-of-the-art (SOTA) pertained language model. From the evolution of the Large language model, plenty of studies have shown tremendous performance in improving NLP tasks. So, I can't agree with calling SciBERT a SOTA model now.
b) The authors failed to mention the hyper-parameters.
c) The authors have failed to show the number of features they selected for the SciBERT to fine-tune.
d) Can regular expression generalize well if the annotated dataset is selected from another population of the collected dataset? This is a concern.
e) Although AI article classification was a case study, it begs the question of reproducibility because of the poor strategy and ambiguous approach of selecting 4,000 research articles.

Validity of the findings

a) I wonder about the performance reported in the original papers for P, R, and F1 scores, especially for Liu's approach to identifying or classifying articles related to emerging fields.
b) I understood that Ensemble AI outperformed the previous method. However, I am unsure if the authors have covered most of the 'emerging field' research articles in the 4,000 annotated data.
c) The novelty of the proposed method has not been established well. See my comprehensive comments in the "Experimental Design."
d) Thesis statement is unclear in the abstract and introduction section. What is the limitation of the proposed method? Do you think there are no limitations?
e) What are the future directions of this research? As a reviewer, I could not identify it.
f) I wonder how current SOTA models related to LLM will classify the research articles in emerging fields.

Annotated reviews are not available for download in order to protect the identity of reviewers who chose to remain anonymous.

Reviewer 3 ·

Basic reporting

no comments

Experimental design

no comments

Validity of the findings

no comments

Additional comments

The paper addressed the issue of poor recall when searching for research articles in emerging fields; however, there was no clear focus on showing that this poor recall is due to the inherent incompleteness of the keyword list. This is especially true as the data size used was the same as that of the compared work. More could have been done to show that the keyword curation was indeed a major reason why the recall was improved.
The paper was able to show the effectiveness of using an ensemble of models to classify research articles, using articles on Artificial Intelligence as a case study. Although the models used were not novel, the ideas presented are novel in that an ensemble of these models presents an interesting use in identifying specific area research articles.



Line 104-105 Inconsistent reference format. This is true for other areas of the paper.

Grammar and Sentence Correction
Line 92-93 Finally we used a Support Vector Machine (SVM) to merge the results from the decision tree, SciBERT and keyword matching to provide a final classification.
Line 95-97 We manually labelled a set of 4000 articles as either AI-related or non-AI for training, and these labels were used to calculate the precision and recall of our approach by 5-fold cross-validation. The results yielded a precision of 92% and a recall of 97%.
Line 226-228 As mentioned above, Liu et al.(2021a) proposed a comprehensive search strategy for AI-related articles with widely used results. This consists of a core lexical query, two expanded lexical queries, and the WoS category “Artificial Intelligence”
Line 251 articles → article

Clarification
Line 144-146 Are these ML methods better at capturing AI innovation's complex and evolving nature or AI-related articles?
Line 153-154 This sentence is not clear enough. What models?
Line 232 it is not so clear what Tang’s approach means
Line 296 A specific figure would have helped bolster this point better.
Line 309 C_x was not defined
Figure 3 is not clear enough about what information the figure is trying to convey
Line 432-434 “Dunham et al. (2020) estimated their approach…”. Whose approach, Dunham’s or Siebert’s? It is not clear enough whose approach is being discussed here.
Line 436 What reported values?

---

## Round 0.2 · Minor Revisions

The authors have addressed most concerns raised by the reviewers. However, some minor/formatting revisions remain. I encourage the authors to address these.

Reviewer 1 ·

Basic reporting

The authors have effectively made revisions based on most of the recommendations I have suggested. However, there are still some formatting issues. For instance, the opening quotation marks for all single and double quotations need to be corrected. Other than that, the authors have fixed most of the basic reporting issues.

Experimental design

As suggested in my previous review, the authors included a detailed explanation of how they methodically selected decision tree and SVM techniques from other machine learning options.

Validity of the findings

No comment.

Additional comments

I would recommend the authors to fix the minor formatting issues that I mentioned in the basic reporting section.

Reviewer 3 ·

Basic reporting

no comment

Experimental design

no comment

Validity of the findings

no comment

Additional comments

Line 104-105 Inconsistent reference format.
Line 136-134 Inconsistent reference format.
Line 140 Inconsistent reference format.
Line 145 Inconsistent reference format.
Line 226 Inconsistent reference format.




Grammar and Sentence Correction
Line 56-57 However, accurate identification within the vast corpus of scientific literature is challenging due to AI’s broad, fuzzy, and rapidly-changing nature(WIPO, 2019).

Clarification
Line 345-348 The simplest approach of using a single keyword “Artificial Intelligence” as the query yielded a very low recall, therefore it is omitted in the following analysis. Can you clarify what this simplest approach is? Maybe making a reference to when it was discussed will give the clarification.

---

## Round 0.3 · accepted · Accept

The authors have addressed the concerns of the reviewers, and the manuscript is ready for publication pending minor edits related to citations as I detail below.

Citations:
Ensure to add a single space before your citations for your final submission.

E.g.,
line 27: science(White, 2019) --> science (White, 2019)
line 28: evolution(WIPO, 2019) --> evolution (WIPO, 2019)
etc.